# PARAMETRIC ADVERSARIAL DIVERGENCES ARE GOOD TASK LOSSES FOR GENERATIVE MODELING

## ABSTRACT

Generative modeling of high dimensional data like images is a notoriously difficult and ill-defined problem. In particular, how to evaluate a learned generative model is unclear. In this paper, we argue that *adversarial learning*, pioneered with generative adversarial networks (GANs), provides an interesting framework to implicitly define more meaningful task losses for unsupervised tasks, such as for generating "visually realistic" images. By relating GANs and structured prediction under the framework of statistical decision theory, we put into light links between recent advances in structured prediction theory and the choice of the divergence in GANs. We argue that the insights about the notions of "hard" and "easy" to learn losses can be analogously extended to adversarial divergences. We also discuss the attractive properties of parametric adversarial divergences for generative modeling, and perform experiments to show the importance of choosing a divergence that reflects the final task.

## 1 INTRODUCTION

For structured prediction and data generation the notion of **final task** is at the same time crucial and not well defined. Consider machine translation; the goal is to predict a good translation, but even humans might disagree on the correct translation of a sentence. Moreover, even if we settle on a ground truth, it is hard to define what it means for a candidate translation to be close to the ground truth. In the same way, for data generation, the task of generating pretty pictures or more generally realistic samples is not well defined. Nevertheless, both for structured prediction and data generation, we can try to define **criteria** which characterize good solutions such as grammatical correctness for translation or non-blurry pictures for image generation. By incorporating enough criteria into a **task loss**, one can hope to approximate the final task, which is otherwise hard to formalize.

Supervised learning and structured prediction are well-defined problems once they are formulated as the minimization of such a task loss. The usual task loss in object classification is the generalization error associated with the classification error, or 0-1 loss. In machine translation, where the goal is to predict a sentence, a **structured loss**, such as the BLEU score (Papineni et al., 2002), formally specifies how close the predicted sentence is from the ground truth. The generalization error is defined through this structured loss. In both cases, models can be objectively compared and evaluated with respect to the task loss (i.e., generalization error). On the other hand, we will show that it is not as obvious in generative modeling to define a task loss that correlates well with the final task of generating realistic samples.

Traditionally in statistics, distribution learning is formulated as density estimation where the task loss is the expected negative-log-likelihood. Although log-likelihood works fine in low-dimension, it was shown to have many problems in high-dimension (Arjovsky et al., 2017). Among others, because the Kullback-Leibler is too strong of a divergence, it can easily saturate whenever the distributions are too far apart, which makes it hard to optimize. Additionally, it was shown in Theis et al. (2016) that the KL-divergence is a bad proxy for the visual quality of samples.

In this work we give insights on how adversarial divergences (Liu et al., 2017) can be considered as task losses and how they address some problems of the KL by indirectly incorporating hard-to-define

criteria. We define **parametric adversarial divergences** as the following :

$$\mathrm{Div}_{\mathrm{NN}}(p||q_\theta) \widehat{=} \sup_{\phi \in \Phi} \mathbf{E}_{(\boldsymbol{x},\boldsymbol{x}') \sim p \otimes q_\theta}[\Delta(f_\phi(\boldsymbol{x}), f_\phi(\boldsymbol{x}'))] \qquad (1)$$

where $\{f_\phi : \mathcal{X} \to \mathbb{R}^{d'} ; \phi \in \Phi\}$ is a class of parametrized functions, such as neural networks, called the **discriminators** in the Generative Adversarial Network (GAN) framework (Goodfellow et al., 2014). The constraints $\Phi$ and the function $\Delta : \mathbb{R}^{d'} \times \mathbb{R}^{d'} \to \mathbb{R}$ determine properties of the resulting divergence. Using these notations, we adopt the view[1] that training a GAN can be seen as training a generator network $q_\theta$ (parametrized by $\theta$) to minimize the parametric adversarial divergence $\mathrm{Div}_{\mathrm{NN}}(p||q_\theta)$, where the generator network defines the probability distribution $q_\theta$ over $\boldsymbol{x}$.

Our contributions are the following:

- We show that compared to traditional divergences, parametric adversarial divergences offer a good compromise in terms of sample complexity, computation, ability to integrate prior knowledge, flexibility and ease of optimization.

- We relate structured prediction and generative adversarial networks using statistical decision theory, and argue that they both can be viewed as formalizing a final task into the minimization of a statistical task loss.

- We explain why it is necessary to choose a divergence that adequately reflects our final task in generative modeling. We make a parallel with results in structured learning (also dealing with high-dimensional data), which quantify the importance of choosing a good objective in a specific setting.

- We explore with some simple experiments how the properties of the discriminator transfer to the adversarial divergence. Our experiments suggest that parametric adversarial divergences are especially adapted to problems such as image generation, where it is hard to formally define a perceptual loss that correlates well with human judgment.

- We illustrate the importance of having a parametric discriminator by running experiments with the true (nonparametric) Wasserstein, and showing its shortcomings on complex datasets, on which GANs are known to perform well.

- We perform qualitative and quantitative experiments to compare maximum-likelihood and parametric adversarial divergences under two settings: very high-dimensional images, and learning data with specific constraints.

## 2    BACKGROUND

Here we briefly introduce the structured prediction framework because it can be related to generative modeling in some ways. We will later link them formally, and present insights from recent theoretical results to choose a better divergence. We also unify parametric adversarial divergences with traditional divergences in order to compare them in the next section.

### 2.1    STRUCTURED PREDICTION

The goal of structured prediction is to learn a classifier $h_\theta : \mathcal{X} \to \mathcal{Y}$ which predicts a structured output $\boldsymbol{y}$ from an input $\boldsymbol{x}$. The key difficulty is that $\mathcal{Y}$ usually has size exponential in the input[2] (e.g. it could be all possible sequence of symbols with a given length). Being able to handle this exponentially large set of outputs is one of the key challenges in structured prediction because it makes traditional multi-class classification methods unusable in general.[3] Standard practice in structured prediction (Taskar et al., 2003; Collins, 2002; Pires et al., 2013) is to consider predictors based on score functions $h_\theta(\boldsymbol{x}) \widehat{=} \arg\max_{\boldsymbol{y}' \in \mathcal{Y}} s_\theta(\boldsymbol{x}, \boldsymbol{y}')$, where $s_\theta : \mathcal{X} \times \mathcal{Y} \to \mathbb{R}$, called the **score/energy function** (LeCun et al., 2006), assigns a score to each possible label $\boldsymbol{y}$ for an input $\boldsymbol{x}$. Typically,

---

[1]We focus in this paper on the divergence minimization perspective of GANs. There are other views, such as those based on game theory (Arora et al., 2017), ratio matching and moment matching (Mohamed & Lakshminarayanan, 2016).

[2]Additionally, $\mathcal{Y}$ might depend on the input $x$, but we ignore this effect for clarity of exposition.

[3]Such as ones based on maximum likelihood.

as in structured SVMs (Taskar et al., 2003), the score function is linear: $s_\theta(\boldsymbol{x}, \boldsymbol{y}) = \langle \theta, g(\boldsymbol{x}, \boldsymbol{y}) \rangle$, where $g(\cdot)$ is a predefined feature map. Alternatively, the score function could also be a learned neural network (Belanger & McCallum, 2016).

In order to evaluate the predictions objectively, we need to define a **task-dependent** structured loss $\ell(\boldsymbol{y'}, \boldsymbol{y}\,;\boldsymbol{x})$ which expresses the cost of predicting $\boldsymbol{y'}$ for $\boldsymbol{x}$ when the ground truth is $\boldsymbol{y}$. We discuss the relation between the loss function and the actual final task in Section 4.2 . The goal is then to find a parameter $\theta$ which minimizes the generalization error:

$$\min_{\theta \in \Theta} \mathbf{E}_{(\boldsymbol{x}, \boldsymbol{y}) \sim p} \left[ \ell(h_\theta(\boldsymbol{x}), \boldsymbol{y}, \boldsymbol{x}) \right] \tag{2}$$

Directly minimizing (2) is often an intractable problem; this is the case when the structured loss $\ell$ is the 0-1 loss (Arora et al., 1993). Instead, the usual practice is to minimize a surrogate loss $\mathbf{E}_{(\boldsymbol{x}, \boldsymbol{y}) \sim p} \left[ \mathcal{L}(s_\theta(\boldsymbol{x}, \boldsymbol{y}), \boldsymbol{y}, \boldsymbol{x}) \right]$ (Bartlett et al., 2006) which has nicer properties, such as sub-differentiability or convexity, to get a tractable optimization problem. The surrogate loss is said to be consistent (Osokin et al., 2017) when its minimizer is also a minimizer of the task loss.

A simple example of structured prediction task is machine translation. Suppose we want to translate French sentences to English; the input $\boldsymbol{x}$ is then a sequence of French words, and the output $\boldsymbol{y}$ is a sequence of English words belonging to a dictionary $D$ with typically $|D| \approx 10000$ words. If we restrict the output sequence to be shorter than $T$ words, then $|\mathcal{Y}| = |D|^T$, which is exponential. An example of desirable criterion is to have a translation with many words in common with the ground truth, which is typically enforced using BLEU scores to define the task loss.

## 2.2 Adversarial and Traditional Divergences

Because we will compare properties of adversarial and traditional divergences throughout this paper, we choose to first unify them with a formalism similar to Sriperumbudur et al. (2012); Liu et al. (2017):

$$\mathrm{Div}(p||q_\theta) \widehat{=} \sup_{f \in \mathcal{F}} \mathbf{E}_{(\boldsymbol{x}, \boldsymbol{x'}) \sim p \otimes q_\theta} [\Delta(f(\boldsymbol{x}), f(\boldsymbol{x'}))] \tag{3}$$

Under this framework we give some examples of **traditional nonparametric divergences**:

- $\psi$-divergences with generator function $\psi$ (which we call f-divergences) can be written in dual form (Nowozin et al., 2016)[4]

$$\mathrm{Div}_\psi(p||q_\theta) \widehat{=} \sup_{f : \mathcal{X} \to \mathbb{R}} \mathbf{E}_{\boldsymbol{x} \sim p}[f(\boldsymbol{x})] - \mathbf{E}_{\boldsymbol{x'} \sim q_\theta}[\psi^*(f(\boldsymbol{x'}))] \tag{4}$$

  where $\psi^*$ is the convex conjugate. Depending on $\psi$, one can obtain any $\psi$-divergence such as the (reverse) Kullback-Leibler, the Jensen-Shannon, the Total Variation, the Chi-Squared[5].

- Wasserstein-1 distance induced by an arbitrary norm $\|\cdot\|$ and its corresponding dual norm $\|\cdot\|^*$ (Sriperumbudur et al., 2012):

$$W(p||q_\theta) \widehat{=} \sup_{\substack{f : \mathcal{X} \to \mathbb{R} \\ \forall \boldsymbol{x} \in \mathcal{X}, \\ \|f'(\boldsymbol{x})\|^* \le 1}} \mathbf{E}_{\boldsymbol{x} \sim p}[f(\boldsymbol{x})] - \mathbf{E}_{\boldsymbol{x'} \sim q_\theta}[f(\boldsymbol{x'})] \tag{5}$$

  which can be interpreted as the cost to transport all probability mass of $p$ into $q$, where $\|\boldsymbol{x} - \boldsymbol{x'}\|$ is the unit cost of transporting $\boldsymbol{x}$ to $\boldsymbol{x'}$.

- Maximum Mean Discrepancy (Gretton et al., 2012):

$$\mathrm{MMD}(p||q_\theta) \widehat{=} \sup_{\substack{f \in \mathcal{H} \\ \|f\|_\mathcal{H} \le 1}} \mathbf{E}_{\boldsymbol{x} \sim p}[f(\boldsymbol{x})] - \mathbf{E}_{\boldsymbol{x'} \sim q_\theta}[f(\boldsymbol{x'})] \tag{6}$$

  where $(\mathcal{H}, K)$ is a Reproducing Kernel Hilbert Space induced by a Kernel $K(\boldsymbol{x}, \boldsymbol{x'})$ on $\mathcal{X}$ with the associated norm $\|\cdot\|_\mathcal{H}$. The MMD has many interpretations in terms of moment-matching (Li et al., 2017).

---

[4]The standard form is $\mathbf{E}_{\boldsymbol{x} \sim q_\theta}[\psi(\frac{p(x)}{q_\theta(x)})]$.

[5]For instance the Kullback-Leibler $\mathbf{E}_{x \sim p}[\log \frac{p(x)}{q_\theta(x)}]$ has the dual form $\sup_{f : \mathcal{X} \to \mathbb{R}} \mathbf{E}_{\boldsymbol{x} \sim p}[f(\boldsymbol{x})] - \mathbf{E}_{\boldsymbol{x'} \sim q_\theta}[\exp(f(\boldsymbol{x'}) - 1)]$. Some $\psi$ require additional constraints, such as $\|f\|_\infty \le 1$ for the Total Variation.

| Divergence | Sample Complexity | Computation | Can Integrate Final Task? |
|---|---|---|---|
| f-Div (explicit model) | $O(1/\epsilon^2)$ | Monte-Carlo, $O(n)$ | No |
| f-Div (implicit model) | N/A | N/A | N/A |
| Nonparametric Wasserstein | $O(1/\epsilon^{d+1})$ | Sinkhorn, $O(n^2)$ | in base distance |
| MMD | $O(1/\epsilon^2)$ | analytic, $O(n^2)$ | in kernel |
| Parametric Adversarial | $O(p/\epsilon^2)$ | SGD | in discriminator |
| Parametric Wasserstein | $O(p/\epsilon^2)$ | SGD | in discriminator & base distance |

Table 1: Properties of Divergences. Explicit and Implicit models refer to whether the density $q_\theta(x)$ can be computed. $p$ is the number of parameters of the parametric discriminator. Sample complexity and computational cost are defined and discussed in Section 3.1, while the ability to integrate desirable properties of the final loss is discussed in Section 3.2. Although f-divergences can be estimated with Monte-Carlo for explicit models, they cannot be easily computed for implicit models without additional assumptions (see text). Additionally, by design, they cannot integrate a final loss directly. The nonparametric Wasserstein can be computed iteratively with the Sinkhorn algorithm, and can integrate the final loss in its base distance, but requires exponentially many samples to estimate. Maximum Mean Discrepancy has good sample complexity, can be estimated analytically, and can integrate the final loss in its base distance, but it is known to lack discriminative power for generic kernels, as discussed below. Parametric adversarial divergences have reasonable sample complexities, can be computed iteratively with SGD, and can integrate the final loss in the choice of class of discriminators. In particular, the parametric Wasserstein has the additional possibility of integrating the final loss into the base distance.

In the optimization problems (4) and (5), whenever $f$ is additionally constrained to be in a given parametric family, the associated divergence will be termed a **parametric adversarial divergence**. In practice, that family will typically be specified as a neural network architecture, so in this work we will use the term **neural adversarial divergences** interchangeably with the slightly more generic **parametric adversarial divergence**. For instance, the parametric adversarial Jensen-Shannon optimized in GANs corresponds to (4) with specific $\psi$ (Nowozin et al., 2016), while the parametric adversarial Wasserstein optimized in WGANs corresponds to (5) where $f$ is a neural network. See Liu et al. (2017) for interpretations and a review and interpretation of other divergences like the Wasserstein with entropic smoothing (Aude et al., 2016), energy-based distances (Li et al., 2017) which can be seen as adversarial MMD, and the WGAN-GP (Gulrajani et al., 2017) objective.

# 3 ADVANTAGES OF PARAMETRIC ADVERSARIAL DIVERGENCES

We argue that parametric adversarial divergences have many good properties which make them attractive for generative modeling. In this section, we compare them to traditional divergences in terms of sample complexity and computational cost (Section 3.1), and ability to integrate criteria related to the final task (Section 3.2). We also discuss the shortcomings of combining the KL-divergence with generators that have a special structure in Section 3.3. We refer the reader to the Appendix for additional interesting properties of parametric adversarial divergences: the optimization and stability issues are discussed in Appendix A.1, the fact that parametric adversarial divergences only make the assumption that one can sample from the generative model, and provide useful learning signal even when their nonparametric counterparts are not well-defined, is discussed in Appendix A.2.

## 3.1 SAMPLE COMPLEXITY AND COMPUTATIONAL COST

Since we want to learn from finite data, we would like to know how well empirical estimates of a divergence approximate the population divergence. In other words, we want to control the **sample complexity**, that is, how many samples $n$ do we need to have with high probability that $|\text{Div}(p||q) - \text{Div}(\widehat{p}_n||\widehat{q}_n)| \leq \epsilon$, where $\epsilon > 0$, and $\widehat{p}_n, \widehat{q}_n$ are empirical distributions associated with $p, q$. Sample complexities for adversarial and traditional divergences are summarized in **Table 1**.

For explicit models which allow evaluating the density $q_\theta(x)$, one could use Monte-Carlo to evaluate the f-divergence with sample complexity $n = O(1/\epsilon^2)$, according to the Central-Limit theorem. For implicit models, there is no one good way of estimating f-divergences from samples. There are some techniques for it (Nguyen et al., 2010; Moon & Hero, 2014; Ruderman et al., 2012), but they all make additional assumptions about the underlying densities (such as smoothness), or they solve the dual in a restricted family, such as a RKHS, which makes the divergences no longer f-divergences.

Parametric adversarial divergences can be formulated as a classification/regression problem with a loss depending on the specific adversarial divergence. Therefore, they have a reasonable sample complexity of $O(p/\epsilon^2)$, where $p$ is the VC-dimension/number of parameters of the discriminator (Arora et al., 2017), and can be solved using classic stochastic gradient methods.

A straightforward nonparametric estimator of the Wasserstein is simply the Wasserstein distance between the empirical distributions $\widehat{p}_n$ and $\widehat{q}_n$, for which smoothed versions can be computed in $O(n^2)$ using specialized algorithms such as Sinkhorn's algorithm (Cuturi, 2013) or iterative Bregman projections (Benamou et al., 2015). However, this empirical Wasserstein estimator has sample complexity $n = O(1/\epsilon^{d+1})$ which is exponential in the number of dimensions (see Sriperumbudur et al., 2012, Corollary 3.5). Thus the empirical Wasserstein is not a viable estimator in high-dimensions.

Maximum Mean Discrepancy admits an estimator with sample complexity $n = O(1/\epsilon^2)$, which can be computed analytically in $O(n^2)$. More details are given in the original MMD paper (Gretton et al., 2007). One should note that MMD depends fundamentally on the choice of kernel. As the sample complexity is independent of the dimension of the data, one might believe that the MMD estimator behaves well in high dimensions. However, it was experimentally illustrated in Dziugaite et al. (2015) that with generic kernels like RBF, MMD performs poorly for MNIST and Toronto face datasets, as the generated images have many artifacts and are clearly distinguishable from the training dataset. See Section 3.2 for more details on the choice of kernel. It was also shown theoretically in (Reddi et al., 2015) that the power of the MMD statistical test can drop polynomially with increasing dimension, which means that with generic kernels, MMD might be unable to discriminate well between high-dimensional generated and training distributions.

Note that comparing divergences in terms of sample complexity can give good insights on what is a good divergence, but should be taken with a grain of salt as well. On the one hand, the sample complexities we give are upper-bounds, which means the estimators could potentially converge faster. On the other hand, one might not need a very good estimator of the divergence in order to learn in some cases. This is illustrated in our experiments with the empirical Wasserstein (Section 6) which has bad sample complexity but yields reasonable results.

### 3.2 Ability to Integrate Desirable Properties for the Final Task

In Section 4, we will argue that in structured prediction, optimizing for the right task losses is more meaningful and can make learning considerably easier. Similarly in generative modeling, we would like divergences to integrate criteria that characterize the final task. We discuss that although not all divergences can easily integrate final task-related criteria, adversarial divergences provide a way to do so.

Pure f-divergences cannot directly integrate any notion of final task,[6] at least not without tweaking the generator. The Wasserstein distance and MMD are respectively induced by a base metric $d(\boldsymbol{x}, \boldsymbol{x}')$ and a kernel $K(\boldsymbol{x}, \boldsymbol{x}')$. The metric and kernel give us the opportunity to specify a task by letting us express a (subjective) notion of similarity. However, the metric and kernel generally have to be defined by hand, as there is no obvious way to learn them end-to-end. For instance, Genevay et al. (2017) learn to generate MNIST by minimizing a smooth Wasserstein based on the L2-distance, while Dziugaite et al. (2015); Li et al. (2015) also learn to generate MNIST by minimizing the MMD induced by kernels obtained externally: either generic kernels based on the L2-distance or on autoencoder features. However, the results seems to be limited to simple datasets. Recently there has been a surge of interest in combining MMD with kernel learning, with convincing results on LSUN, CelebA and ImageNet images. Mroueh et al. (2017) learn a feature map and try to match its mean and covariance, Li et al. (2017) learn kernels end-to-end, while Bellemare et al. (2017) do end-to-end learning of energy distances, which are closely related to MMD.

Parametric adversarial divergences are defined with respect to a parametrized class of discriminators, thus changing properties of the discriminator is a primary way to affect the associated divergence. The form of the discriminator may determine what aspects the divergence will be sensitive or blind to. For instance using a convolutional network as the discriminator may render the divergence insen-

---

[6]One could also attempt to induce properties of interest by adding a regularization term to the f-divergence. However, if we assume that maximum likelihood is itself often not a meaningful task loss, then there is no guarantee that minimizing a tradeoff between maximum likelihood and a regularization term is more meaningful or easier.

sitive to small image translations. Additionally, the parametric adversarial Wasserstein distance (Arjovsky et al., 2017) can also incorporate a custom metric. In Section 6 we give interpretations and experiments to assess the relation between the discriminator and the divergence.

### 3.3 COMBINING KL-DIVERGENCE WITH GENERATORS THAT HAVE SPECIAL STRUCTURE CREATES OTHER PROBLEMS

In some cases, imposing a certain structure on the generator (e.g. a Gaussian or Laplacian observation model) yields a Kullback-Leibler divergence which involves some form of component-wise distance between samples, reminiscent of the Hamming loss (see Section 4.3) used in structured prediction. However, doing maximum likelihood on generators having an imposed special structure can have drawbacks which we detail here. For instance, the generative model of a typical variational autoencoder can be seen as an infinite mixture of Gaussians (Kingma & Welling, 2014). The log-likelihood thus involves a "reconstruction loss", a pixel-wise L2 distance between images analogous to the Hamming loss, which makes the training relatively easy and very stable. However, the Gaussian is partly responsible for the VAE's inability to learn sharp distributions. Indeed it is a known problem that VAEs produce blurry samples (Arjovsky et al., 2017), in fact even if the approximate posterior matches exactly the true posterior, which would correspond to the evidence lower-bound being tight, the output of the VAE would still be blurry (Bousquet et al., 2017). Other examples are autoregressive models such as recurrent neural networks (Mikolov et al., 2010) which factorize naturally as $\log q_\theta(x) = \sum_i \log q_\theta(x_i|x_1, .., x_{i-1})$, and PixelCNNs (Oord et al., 2016). Training autoregressive models using maximum likelihood results in teacher-forcing (Lamb et al., 2016): each ground-truth symbol is fed to the RNN, which then has to maximize the likelihood of the next symbol. Since teacher-forcing induces a lot of supervision, it is possible to learn using maximum-likelihood. Once again, there are similarities with the Hamming loss because each predicted symbol is compared with its associated ground truth symbol. However, among other problems, there is a discrepancy between training and generation. Sampling from $q_\theta$ would require iteratively sampling each symbol and feeding it back to the RNN, giving the potential to accumulate errors, which is not something that is accounted for during training. See Leblond et al. (2017) and references therein for more principled approaches to sequence prediction with autoregressive models.

## 4 CHOOSING BETTER TASK LOSSES

In this section, we try to provide insights in order to design the best adversarial divergence for our final task. After establishing the relationship between structured prediction and generative adversarial networks, we review theoretical results on the choice of objectives in structured prediction, and discuss their interpretation in generative modeling.

### 4.1 RELATING STRUCTURED PREDICTION AND GENERATIVE ADVERSARIAL NETWORKS

We frame the relationship of structured prediction and GANs using the framework of **statistical decision theory**. Assume that we are in a world with a set $\mathcal{P}$ of possible states and that we have a set $\mathcal{A}$ of actions. When the world is in the state $p \in \mathcal{P}$, the cost of playing action $a \in \mathcal{A}$ is the **(statistical) task loss** $L_p(a)$. The goal is to play the action minimizing the task loss.

**Generative models with Maximum Likelihood.** The set $\mathcal{P}$ of possible states is the set of available distributions $\{p\}$ for the data $\boldsymbol{x}$. The set of actions $\mathcal{A}$ is the set of possible distributions$\{q_\theta \; ; \; \theta \in \Theta\}$ for the model and the task loss is the negative log-likelihood,

$$L_p(\theta) \widehat{=} \mathbf{E}_{\boldsymbol{x} \sim p} \left[ -\log(q_\theta(\boldsymbol{x})) \right] \tag{7}$$

**Structured prediction.** The set $\mathcal{P}$ of possible states is the set of available distribution $\{p\}$ for $(\boldsymbol{x}, \boldsymbol{y})$. The set of actions $\mathcal{A}$ is the set of prediction functions $\{h_\theta \; ; \; \theta \in \Theta\}$ and the task loss is the generalization error:

$$L_p(\theta) \widehat{=} \mathbf{E}_{(\boldsymbol{x}, \boldsymbol{y}) \sim p} \left[ \ell(h_\theta(\boldsymbol{x}), \boldsymbol{y}, \boldsymbol{x}) \right] \tag{8}$$

where $\ell : \mathcal{Y} \times \mathcal{Y} \times \mathcal{X} \to \mathbb{R}$ is a structured loss function.

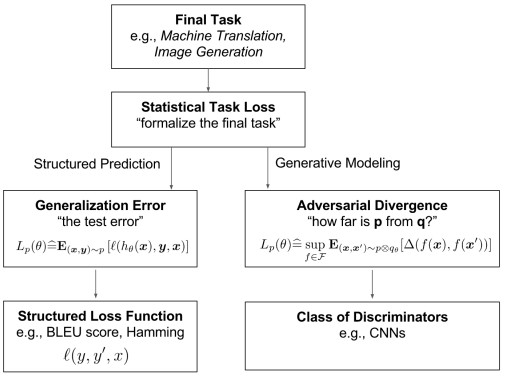

Figure 1: Formalizing a final task into the minimization of a statistical task loss. One starts from a useful but ill-defined final task, and devises criteria that characterize good solutions. Such criteria are integrated into the statistical task loss, which is the generalization error in structured prediction, and the adversarial divergence in the GAN framework. The hope is that minimizing the statistical task loss effectively solves the final task.

**GANs.** The set $\mathcal{P}$ of possible states is the set of available distributions $\{p\}$ for the data $\boldsymbol{x}$. The set of actions $\mathcal{A}$ is the set of distributions $\{q_\theta \; ; \; \theta \in \Theta\}$ that the generator can learn, and the task loss is the adversarial divergence

$$L_p(\theta) \widehat{=} \sup_{f \in \mathcal{F}} \mathbf{E}_{(\boldsymbol{x},\boldsymbol{x}') \sim p \otimes q_\theta}[\Delta(f(\boldsymbol{x}), f(\boldsymbol{x}'))] \tag{9}$$

Under this unified framework, the prediction function $h_\theta$ is analogous to the generative model $q_\theta$, while the choice of the right structured loss $\ell$ can be related to $\Delta$ and to the choice of the discriminator family $\mathcal{F}$ which will induce a good adversarial divergence. We will further develop this analogy in Section 4.2.

## 4.2 Link Between Structured Losses and Adversarial Divergences

As discussed in the introduction, structured prediction and data generation involve a notion of **final task** which is at the same time crucial and not well defined. Nevertheless, for both we can try to define criteria which characterize good solutions. We would like the statistical task loss (introduced in Section 4.1), which corresponds to the generalization error in structured prediction, and the adversarial divergence in generative modeling, to incorporate task-related criteria. One way to do that is to choose a structured loss that reflects the criteria of interest, or analogously to choose a class of discriminators, like a CNN architecture, such that the resulting adversarial divergence has good invariance properties. The whole process of building statistical task losses adapted to a final task, using the right structured losses or discriminators, is represented in Figure 1.

For many prediction problems, the structured prediction community has engineered structured loss functions which induce **properties of interest** on the learned predictors. In machine translation, a commonly considered property of interest is for candidate translations to contain many words in common with the ground-truth; this has given rise to the BLEU score which counts the percentage of candidate words appearing in the ground truth. In the context of image segmentation, Osokin & Kohli (2014) have compared various structured loss functions which induces different properties on the predicted mask.

In the same vein as structured loss functions, adversarial divergences can be built to induce certain properties on the generated data. We are more concerned with generating realistic samples than having samples which are very similar with the training set; we actually want to extrapolate some properties of the true distribution from the training set. For instance, in the DCGAN (Radford et al., 2016), the discriminator has a convolutional architecture, which makes it potentially robust to small deformations that would not affect the visual quality of the samples significantly, while still making it able to detect blurry samples, which is aligned with our objective of generating realistic samples.

## 4.3 "Weaker" Task Losses/Divergences are Easier to Learn

**Intuition on the Flexibility of Losses.** In this section we get insights from the convergence results of Osokin et al. (2017) in structured prediction. They show in a specific setting that some "weaker" structured loss functions are easier to learn than some stronger loss functions. In some sense, their results formalize the intuition in generative modeling that learning with "weaker" divergences is

easier (Arjovsky et al., 2017) and more intuitive (Liu et al., 2017) than stronger divergences. In structured prediction, strong losses such as the 0-1 loss are hard to learn with because they do not give any flexibility on the prediction; the 0-1 loss only tells us whether a prediction is correct or not, and consequently does not give any clue about how close the prediction is to the ground truth. To get enough learning signal, we roughly need as many training examples as the number of possible outputs $|\mathcal{Y}|$, which is exponential in the dimension of $y$ and thus inefficient. Conversely, weaker losses like the Hamming loss have more flexibility; because they tell us how close a prediction is to the ground truth, less examples are needed to generalize well. The theoretical results proved by Osokin et al. (2017) formalize that intuition in a specific setting.

**Theory to Back the Intuition.** In a non-parametric setting (details and limitations in Appendix B), Osokin et al. (2017) formalize the intuition that weaker structured loss functions are easier to optimize. Specifically, they compare the 0-1 loss $\ell_{0-1}(\boldsymbol{y}, \boldsymbol{y}') \widehat{=} \mathbf{1}\{\boldsymbol{y} \neq \boldsymbol{y}'\}$ to the Hamming loss $\ell_{Ham}(\boldsymbol{y}, \boldsymbol{y}') \widehat{=} \frac{1}{T} \sum_{t=1}^{T} \mathbf{1}\{\boldsymbol{y}_t \neq \boldsymbol{y}'_t\}$, when $\boldsymbol{y}$ decomposes as $T = \log_2 |\mathcal{Y}|$ binary variables $(y_t)_{1 \leq t \leq T}$. They derive a worst case sample complexity needed to obtain a fixed error $\epsilon > 0$. For the 0-1 loss, they obtain a sample complexity of $O(|\mathcal{Y}|/\epsilon^2)$ which is exponential in the dimension of $y$. However, for the Hamming loss, under certain constraints (see Osokin et al., 2017, section on exact calibration functions) they obtain a much better sample complexity of $O(\log_2 |\mathcal{Y}|/\epsilon^2)$ which is polynomial in the number of dimensions, whenever certain constraints are imposed on the score function. Thus their results suggest that choosing the right structured loss, like the weaker Hamming loss, might make training exponentially faster.

**Insights and Relation with Adversarial Divergences.** Osokin et al. (2017)'s theoretical results confirm our intuition that weaker losses are easier to optimize, and quantify in a specific setting how much harder it is to learn with strong structured loss functions, like the 0-1 loss, than with weaker ones, like the Hamming loss (here, exponentially harder). Under the framework of statistical decision theory (introduced Section 4.1), their results can be related to analogous results in generative modeling (Arjovsky et al., 2017; Liu et al., 2017) showing that it can be easier to learn with weaker divergences than with stronger ones. In particular, one of their arguments is that distributions with disjoint support can be compared in weaker topologies like the the one induced by the Wasserstein but not in stronger ones like the the one induced by the Jensen-Shannon.

## 5 RELATED WORK

Closest to our work are the following two papers. Arora et al. (2017) argue that analyzing GANs with a nonparametric (optimal discriminator) view does not really make sense, because the usual nonparametric divergences considered have bad sample complexity. They also prove sample complexities for parametric divergences. Liu et al. (2017) prove under some conditions that globally minimizing a neural divergence is equivalent to matching all moments that can be represented within the discriminator family. They unify parametric divergences with nonparametric divergences and introduce the notion of strong and weak divergence. However, both those works do not attempt to study the meaning and practical properties of parametric divergences. In our work, we start by introducing the notion of final task, and then discuss why parametric divergences can be good task losses with respect to usual final tasks. We also perform experiments to determine properties of some parametric divergences, such as invariance, ability to enforce constraints and properties of interest, as well as the difference with their nonparametric counterparts. Finally, we unify structured prediction and generative modeling, which could give a new perspective to the community.

The following papers are also related to our work because of one of the following aspects: unifying divergences, analyzing their statistical properties, giving other interpretations of generative modeling, improving GANs, criticizing maximum-likelihood as a objective for generative modeling, and other reasons. Before the first GAN paper, Sriperumbudur et al. (2012) unify traditional IPMs, analyze their statistical properties, and propose to view them as classification problems. Similarly, Reid & Williamson (2011) show that computing a divergence can be formulated as a classification problem. Later, Nowozin et al. (2016) generalize the GAN objective to any adversarial f-divergence. However, the first papers to actually study the effect of restricting the discriminator to be a neural network instead of any function are the MMD-GAN papers: Li et al. (2015); Dziugaite et al. (2015); Li et al. (2017); Mroueh et al. (2017) and Bellemare et al. (2017) who give an interpretation of their

Figure 2: Images generated by the network after training with the Sinkorn-Autodiff algorithm on MNIST dataset (left) and CIFAR-10 dataset (right). One can observe than although the network succeeds in learning MNIST, it is unable to produce convincing and diverse samples on the more complex CIFAR-10.

energy distance framework in terms of moment matching. Mohamed & Lakshminarayanan (2016) give many interpretations of generative modeling, including moment-matching, divergence minimization, and density ratio matching. On the other hand, work has been done to better understand the GAN objective in order to improve its stability (Salimans et al., 2016). Subsequently, Arjovsky et al. (2017) introduce the adversarial Wasserstein distance which makes training much more stable, and Gulrajani et al. (2017) improve the objective to make it more practical. Regarding model evaluation, Theis et al. (2016) contains an excellent discussion on the evaluation of generative models, they show in particular that log-likelihood is not a good proxy for the visual quality of samples. Danihelka et al. (2017) compare parametric adversarial divergence and likelihood objectives in the special case of RealNVP, a generator with explicit density, and obtain better visual results with the adversarial divergence. Concerning theoretical understanding of learning in structured prediction, some recent papers are devoted to theoretical understanding of structured prediction such as Cortes et al. (2016) and London et al. (2016) which propose generalization error bounds in the same vein as Osokin et al. (2017) but with data dependencies.

One contribution of the present paper is to have taken these results from the prior literature and put them in perspective in an attempt to provide a more principled view of the nature and usefulness of parametric divergences, in comparison to traditional divergences. To the best of our knowledge, we are also the first to make a link between the generalization error of structured prediction and the adversarial divergence in generative modeling.

## 6 EXPERIMENTAL RESULTS

**Importance of Sample Complexity.** Since the sample complexity of the nonparametric Wasserstein is exponential in the dimension (Section 3.1), we check experimentally whether training a generator to minimize the nonparametric Wasserstein distance fails in high dimensions. We implement the Sinkhorn-AutoDiff algorithm (Genevay et al., 2017) to compute the entropy-regularized L2-Wasserstein distance between minibatches of training images and generated images. **Figure 2** shows generated samples after training with the Sinkhorn-Autodiff algorithm on both MNIST and CIFAR-10 dataset. On MNIST, the network manages to produce decent but blurry images. However, on CIFAR-10, which is a much more complex dataset, the network fails to produce meaningful samples, which would suggest that indeed the nonparametric Wasserstein should not be used for generative modeling when the (effective) dimensionality is high. This result is to be contrasted with the recent successes in image generation of the parametric Wasserstein (Gulrajani et al., 2017), which also has much better sample complexity than the nonparametric Wasserstein.

**Robustness to Transformations.** Intuitively, small rotations should not significantly affect the realism of images, while additive noise should. We study the robustness of various parametric adversarial divergences to rotations and additive noise by plotting the evolution of the divergence between MNIST and rotated/noisy versions of it, as a function of the amplitude of transformation. We consider three discriminators (linear, 1-layer-dense, 2-layer-cnn) combined with two formulations, parametric Jensen-Shannon (ParametricJS) and parametric Wasserstein (ParametricW). Ideally, good divergences should vary smoothly (be robust) with respect to the amplitude of the transformation. For rotations (**Figures 3a and 3b**) and all discriminators except the linear, ParametricJS saturates at its maximal value, even for small values of rotation, whereas the Wasserstein distance varies much more smoothly, which is consistent with the example given by Arjovsky et al. (2017). The fact that the linear ParametricJS does not saturate for rotations shows that the architecture of the discriminator has a significant effect on the induced parametric adversarial divergence, and confirms that there is a conceptual difference between the true JS and ParametricJS, and even among

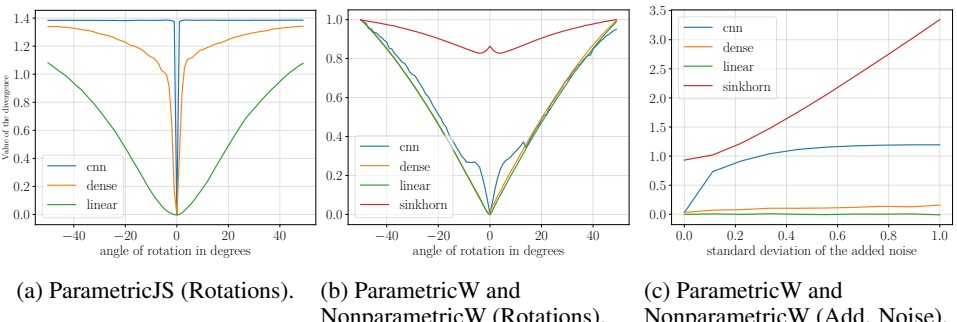

(a) ParametricJS (Rotations).

(b) ParametricW and NonparametricW (Rotations).

(c) ParametricW and NonparametricW (Add. Noise).

Figure 3:   **Top (a) and (b)**: divergences between MNIST and rotated MNIST. **Bottom (c)**: divergences between MNIST and noisy MNIST. NonparametricW was estimated with Sinkhorn's algorithm. ParametricW plots for each model were rescaled, but using the same scaling factor across plots. When comparing different models/divergences, only the shape (but not the scale) of the curve matters, while for a same model the scale across different transformations does matter.

different ParametricJS. For additive Gaussian noise (**Figure 3c**), the linear discriminator is unable to distinguish the two distributions (it only sees the means of the distributions), whereas more complex architectures like CNNs do. In that sense the linear discriminator is too weak for the task, or not strict enough (Liu et al., 2017), which suggests that a better divergence involves trading off between robustness and strength.

**Learning High-dimensional Data.**   We collect *Thin-8*, a dataset of about 1500 handwritten images of the digit "8", with a very high resolution of $512 \times 512$, and augment them with elastic deformations. Because the pen strokes are relatively thin, we expect any pixel-wise distance to be uninformative, because the images are dominated by background pixels, and because with high probability, any two "8' will intersect on no more than a little area. We train a convolutional VAE and a WGAN-GP (Gulrajani et al., 2017), henceforth simply denoted GAN, using nearly the same architectures (VAE decoder similar to GAN generator, VAE encoder similar to GAN discriminator), with 16 latent variables, on the following resolutions: $32 \times 32$, $128 \times 128$ and $512 \times 512$. Generated samples are shown in **Figure 4**. Indeed, we observe that the VAE, trained to minimize the evidence lower bound on maximum-likelihood, fails to generate convincing samples in high-dimensions: they are blurry, pixel values are gray instead of being white, and some samples look like the average of many digits. On the contrary, the GAN can generate sharp and realistic samples even in $512 \times 512$. Our hypothesis is that the discriminator learns moments which are easier to match than it is to directly match the training set with maximum likelihood. Since we were able to perfectly generate high-resolution digits, an additional insight of our experiment is that the main difficulty in generating high-dimensional natural images (like ImageNet and LSUN bedrooms) resides not in high resolution itself, but in the intrinsic complexity of the scenes. Such complexity can be hidden in low resolution, which might explain recent successes in generating images in low resolution but not in higher ones.

**Learning Visual Hyperplanes.**   We design the **visual hyperplane task** to be able to compare VAEs and GANs quantitatively rather than simply inspecting the quality of their generated images. We create a new dataset by concatenating sets of 5 images from MNIST, such that those digits sum up to 25. We train a VAE and a WGAN-GP (henceforth simply denoted GAN) on this new dataset (we used 4504 combinations out of the 5631 possible combinations for training). Both model share the same architecture for generator network and use 200 latent variables. With the help of a MNIST classifier, we automatically recognize and sum up the digits in each generated sample. **Figure 5** shows the distributions of the sums of the digits generated by the VAE and GAN[7]. We can see that the GAN distribution is more peaked and centered around the target 25, while the VAE distribution is less precise and not centered around the target. In that respect, the GAN was better than the VAE at capturing the particular aspects and constraints of the data distribution (summing up to 25). One

_______________

[7]See appendix to view actual sampled images: as usual, the VAE samples are mostly blurry while the GAN samples are more realistic and crisp.

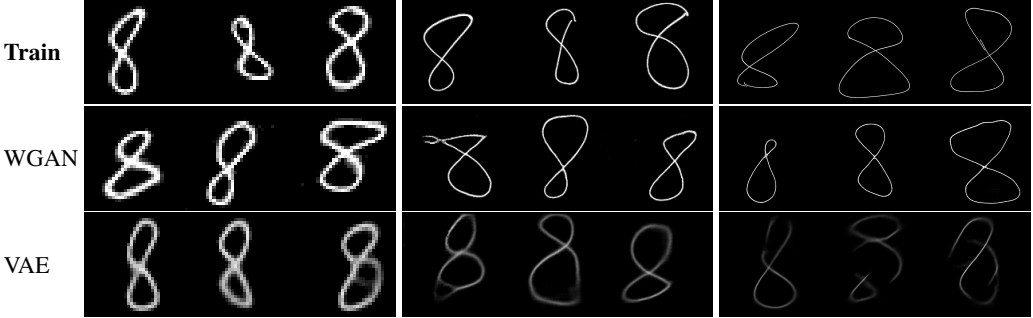

Figure 4: Samples from *Thin-8* training set (**top row**), WGAN-GP (**middle row**) and Convolutional VAE (**bottom row**) with 16 latent variables. Resolutions are $32 \times 32$ (**left column**), $128 \times 128$ (**middle column**), and $512 \times 512$ (**right column**). Note how the GAN samples are always crips and realistic across all resolutions, while the VAE samples tend to be blurry with gray pixel values in high-resolution. We can also observe some averaging artifacts in the top-right 512x512 VAE sample, which looks like the average of two "8". More samples can be found in Section C.2 of the Appendix.

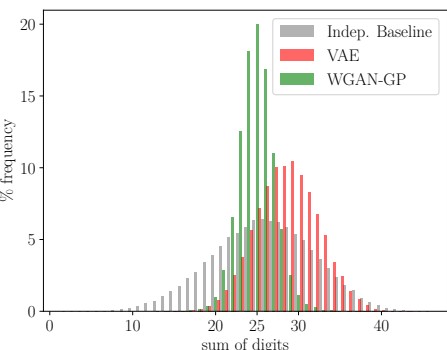

Figure 5: Histograms of the sums of digits generated by VAE (red), WGAN-GP (green) and Independent Baseline (gray). The latter draws digits independently according to their empirical marginal probabilities, which corresponds to fitting independent multinomial distributions over digits using maximum likelihood. WGAN-GP beats largely both VAE and Indepedent Baseline as it gives a sharper distribution centered in the target sum 25.

possible explanation is that since training a classifier to recognize digits and sum them up is not hard in a supervised setting, it could also be relatively easy for a discriminator to enforce such a constraint.

## 7  CONCLUSION

We gave arguments in favor of using adversarial divergences rather than traditional divergences for generative modeling, the most important of which being the ability to account for the final task. After linking structured prediction and generative modeling under the framework of statistical decision theory, we interpreted recent results from structured prediction, and related them to the notions of strong and weak divergences. Moreover, viewing adversarial divergences as statistical task losses led us to believe that some adversarial divergences could be used as evaluation criteria in the future, replacing hand-crafted criteria which cannot usually be exhaustive. In some sense, we want to extrapolate a few desirable properties into a meaningful task loss. In the future we would like to investigate how to define meaningful evaluation criteria with minimal human intervention.

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

## A    ADVANTAGES OF PARAMETRIC ADVERSARIAL DIVERGENCES

In this section, we describe additional advantages and properties of parametric adversarial divergences.

### A.1    EASE OF OPTIMIZATION AND STABILITY

While adversarial divergences are learned and thus potentially much more powerful than traditional divergences, the fact that they are the solution to a hard, non-convex problem can make GANs unstable. Not all adversarial divergences are equally stable: Arjovsky et al. (2017) claimed that the adversarial Wasserstein gives more meaningful learning signal than the adversarial Jensen-Shannon, in the sense that it correlates well with the quality of the samples, and is less prone to mode dropping. In Section 6 we will show experimentally on a simple setting that indeed the neural adversarial Wasserstein consistently give more meaningful learning signal than the neural adversarial Jensen-Shannon, regardless of the discriminator architecture. Similarly to the WGAN, the MMD-GAN divergence (Li et al., 2017) was shown to correlate well with the quality of samples and to be robust to mode collapse. Recently, it was shown that neural adversarial divergences other than the Wasserstein can also be made stable by regularizing the discriminator properly (Kodali et al., 2017; Roth et al., 2017).

### A.2    SAMPLING FROM GENERATOR IS SUFFICIENT

Maximum-likelihood typically requires computing the density $q_\theta(x)$, which is not possible for implicit models such as GANs, from which it is only possible to sample. On the other hand, parametric adversarial divergences can be estimated with reasonable sample complexity (see Section 3.1) only by sampling from the generator, without any assumption on the form of the generator. This is also true for MMD but generally not the case for the empirical Wasserstein, which has bad sample complexity as stated previously. Another issue of f-divergences such as the Kullback-Leibler and the Jensen-Shannon is that they are either not defined (Kullback-Leibler) or uninformative (Jensen-Shannon) when $p$ is not absolutely continuous w.r.t. $q_\theta$ (Nowozin et al., 2016), which makes them unusable for learning sharp distributions such as manifolds. On the other hand, some integral probability metrics, such as the Wasserstein, MMD, or their adversarial counterparts, are well defined for any distributions $p$ and $q_\theta$. In fact, even though the Jensen-Shannon is uninformative for manifolds, the parametric adversarial Jensen-Shannon used in the original GANs (Goodfellow et al., 2014) still allows learning realistic samples, even though the process is unstable (Salimans et al., 2016).

## B    LIMITATIONS OF OSOKIN ET AL. (2017)'S THEORY.

Although Osokin et al. (2017) give a lot of insights, their results must be taken with a grain of salt. In this section we point out the limitations of their theory.

First, their analysis ignores the dependence on $x$ and is non-parametric, which means that they consider the whole class of possible score functions for each given $x$. Additionally, they only consider convex consistent surrogate losses in their analysis, and they give upper bounds but not lower bounds on the sample complexity. It is possible that optimizing approximately-consistent surrogate losses instead of consistent ones, or making additional assumptions on the distribution of the data could yield better sample complexities.

## C  EXPERIMENTAL RESULTS

### C.1  LEARNABILITY OF PARAMETRIC ADVERSARIAL DIVERGENCES.

Here, we compare the parametric adversarial divergences induced by three different discriminators (linear, dense, and CNN) under the WGAN-GP (Gulrajani et al., 2017) formulation.

We consider one of the simplest non-trivial generators, in order to factor out optimization issues on the generator side. The model is a mixture of 100 Gaussians with zero-covariance. The model density is $q_\theta(\boldsymbol{x}) = \frac{1}{K} \sum_z \delta(\boldsymbol{x} - \boldsymbol{x}_z)$, parametrized by prototypes $\theta = (\boldsymbol{x}_z)_{1 \le z \le K}$. The generative process consists in sampling a discrete random variable $z \in \{1, ..., K\}$, and returning the prototype $\boldsymbol{x}_z$.

Learned prototypes (means of each Gaussian) are shown in **Figure 6 and 7**. The first observation is that the linear discriminator is too weak of a divergence: all prototypes only learn the mean of the training set. Now, the dense discriminator learns prototypes which sometimes look like digits, but are blurry or unrecognizable most the time. The samples from the CNN discriminator are never blurry and recognizable in the majority of cases. Our results confirms that indeed, even for simplistic models like a mixture of Gaussians, using a CNN discriminator provides a better task loss for generative modeling of images.

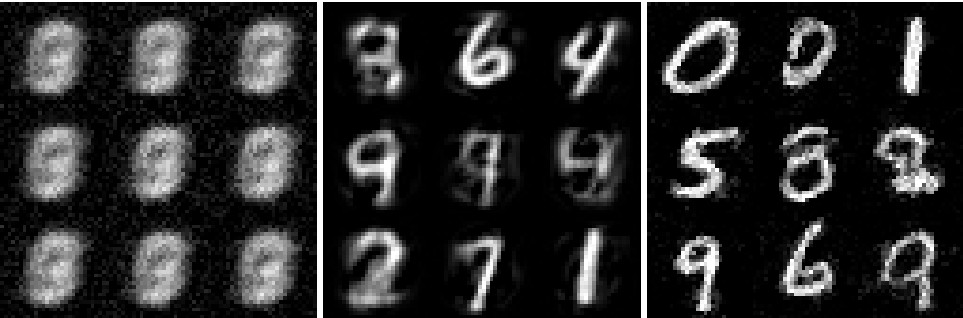

Figure 6: Some Prototypes learned using linear (**left**), dense (**middle**), and CNN discriminator (**right**). We observe that with linear discriminator, only the mean of the training set is learned, while using the dense discriminator yields blurry prototypes. Only using the CNN discriminator yields clear prototypes. All 100 prototypes can be found in Figure 7.

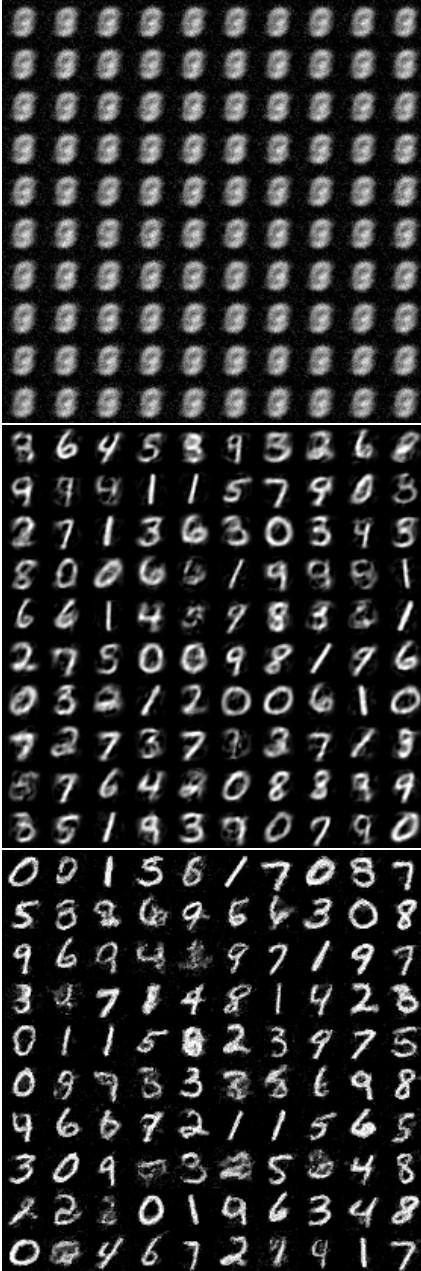

Figure 7: All 100 Prototypes learned using linear (**top**), dense (**middle**), and CNN discriminator (**bottom**). We observe that with linear discriminator, only the mean of the training set is learned, while using the dense discriminator yields blurry prototypes. Only the CNN discriminator yields clear prototypes.

## C.2 ADDITIONAL SAMPLES FOR VAE AND GAN

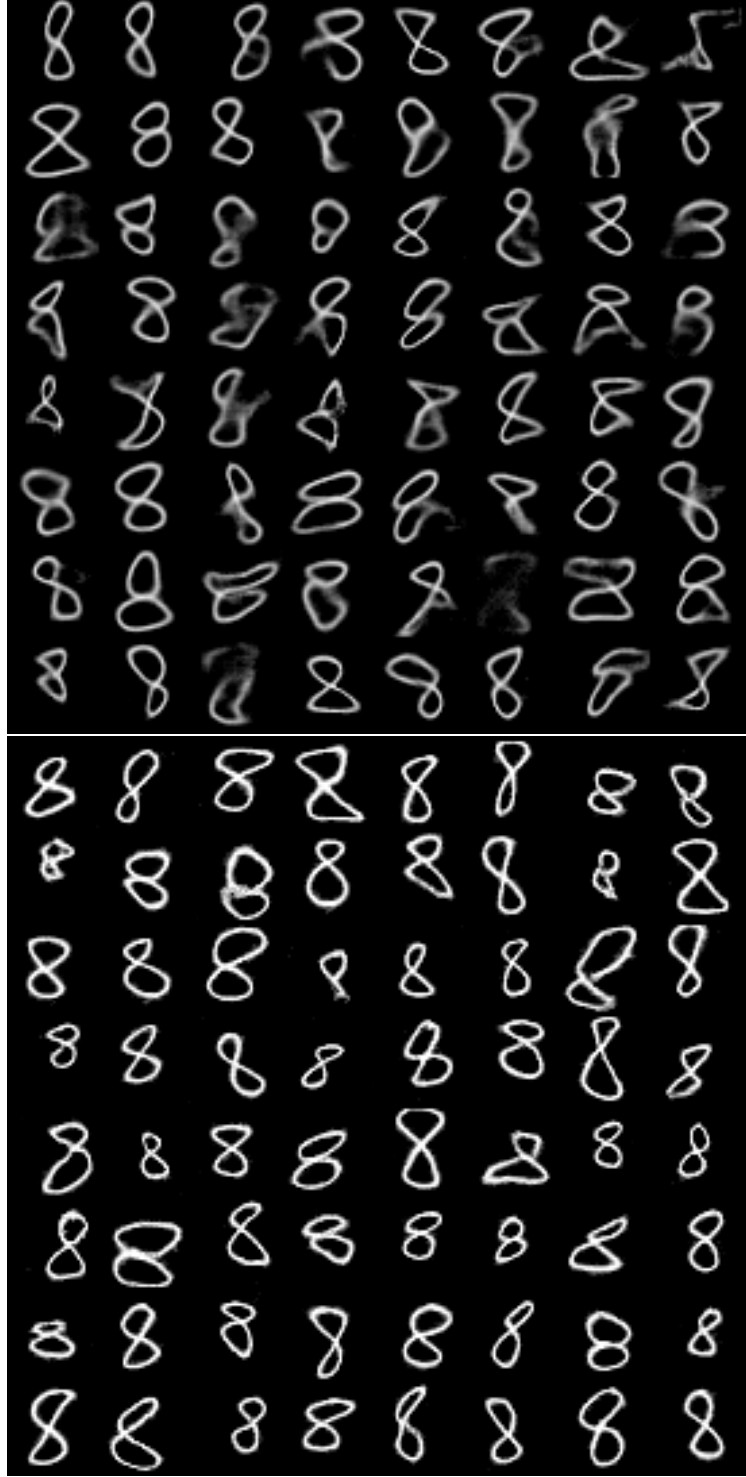

Figure 8: VAE (**top**) and GAN (**bottom**) samples with 16 latent variables and $32 \times 32$ resolution.

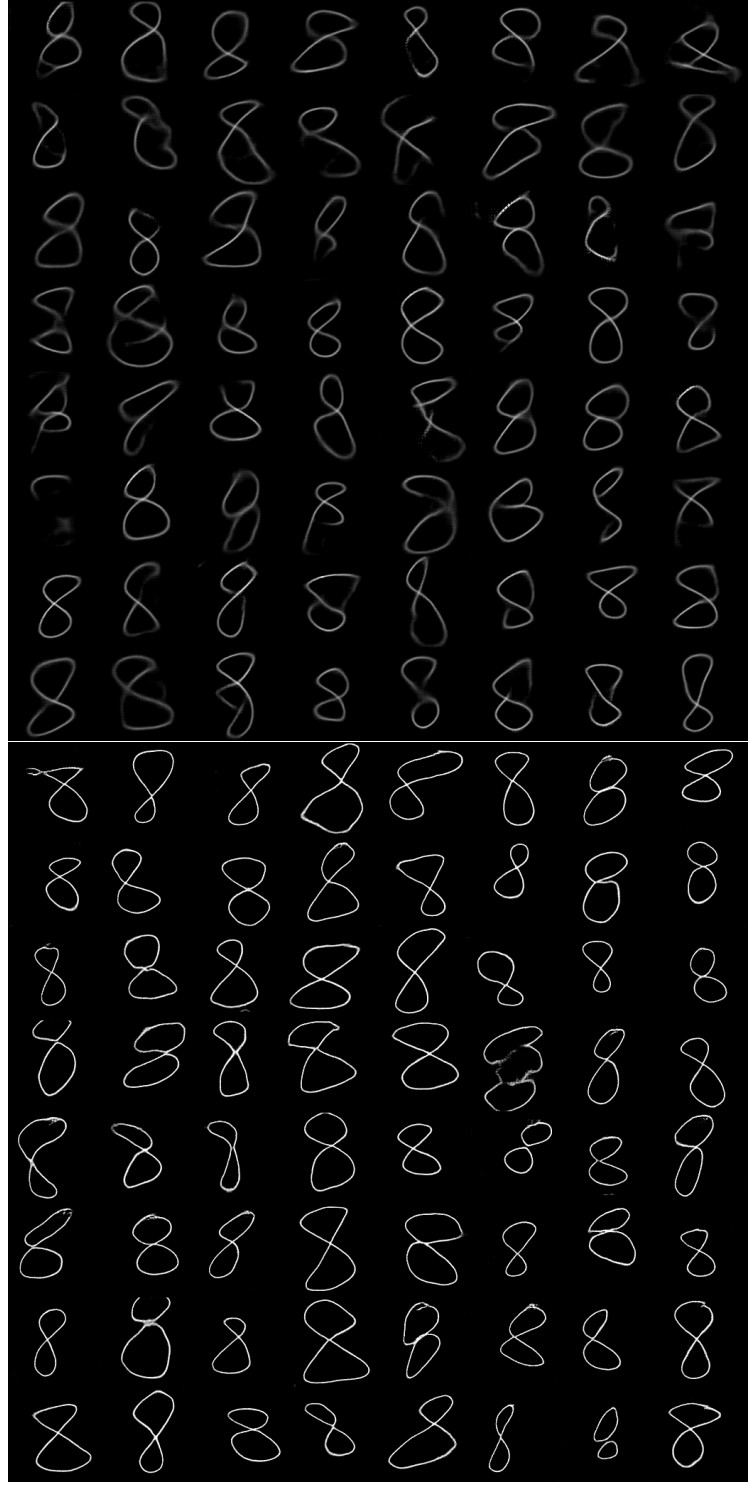

Figure 9: VAE (**top**) and GAN (**bottom**) samples with 16 latent variables and $128 \times 128$ resolution.

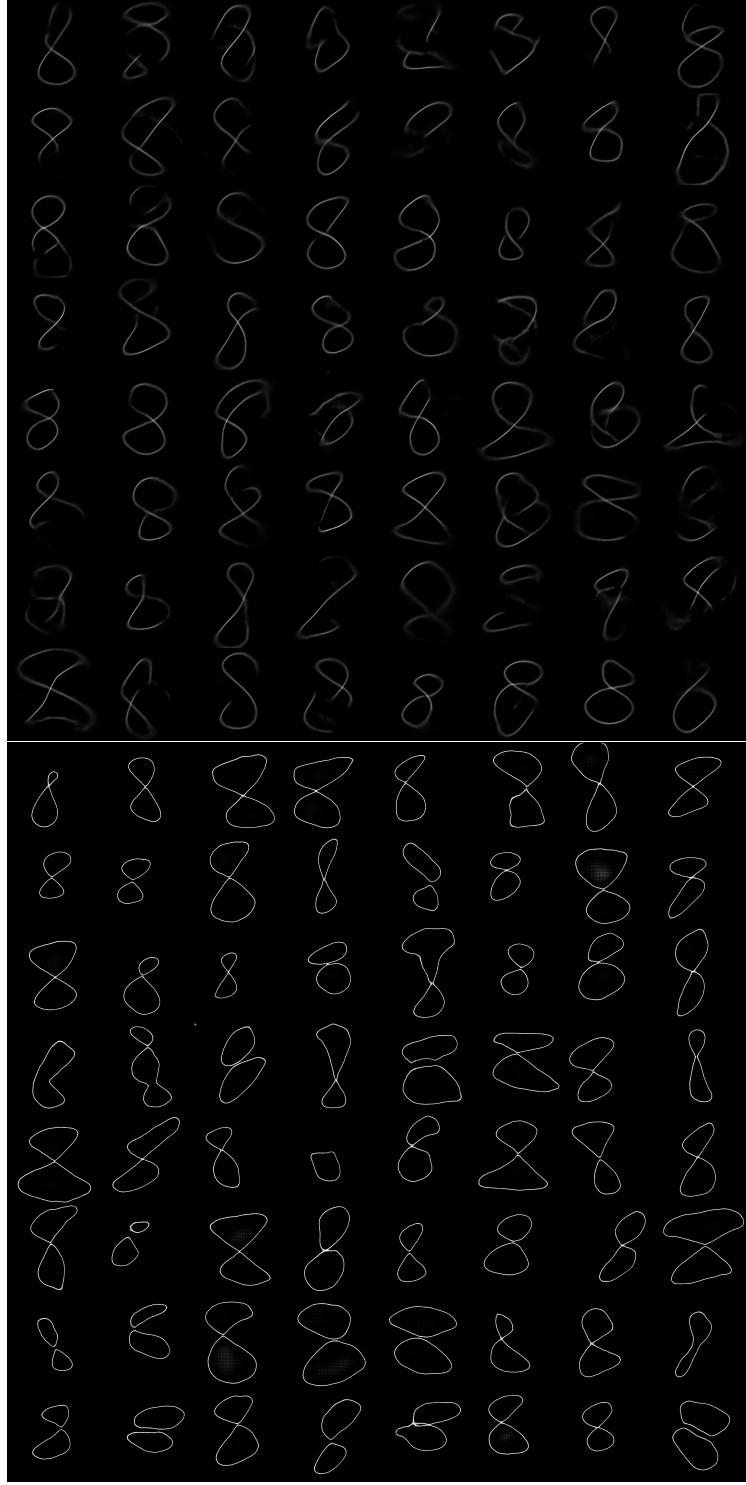

Figure 10: VAE (**top**) and GAN (**bottom**) samples with 16 latent variables and $512 \times 512$ resolution.

### C.3 VISUAL HYPERPLANE: ADDITIONAL SAMPLES

Figure 11 shows some additional samples from the VAE and WGAN-GP trained on the visual-hyperplane task. Both models have 200 latent variables and similar architectures.

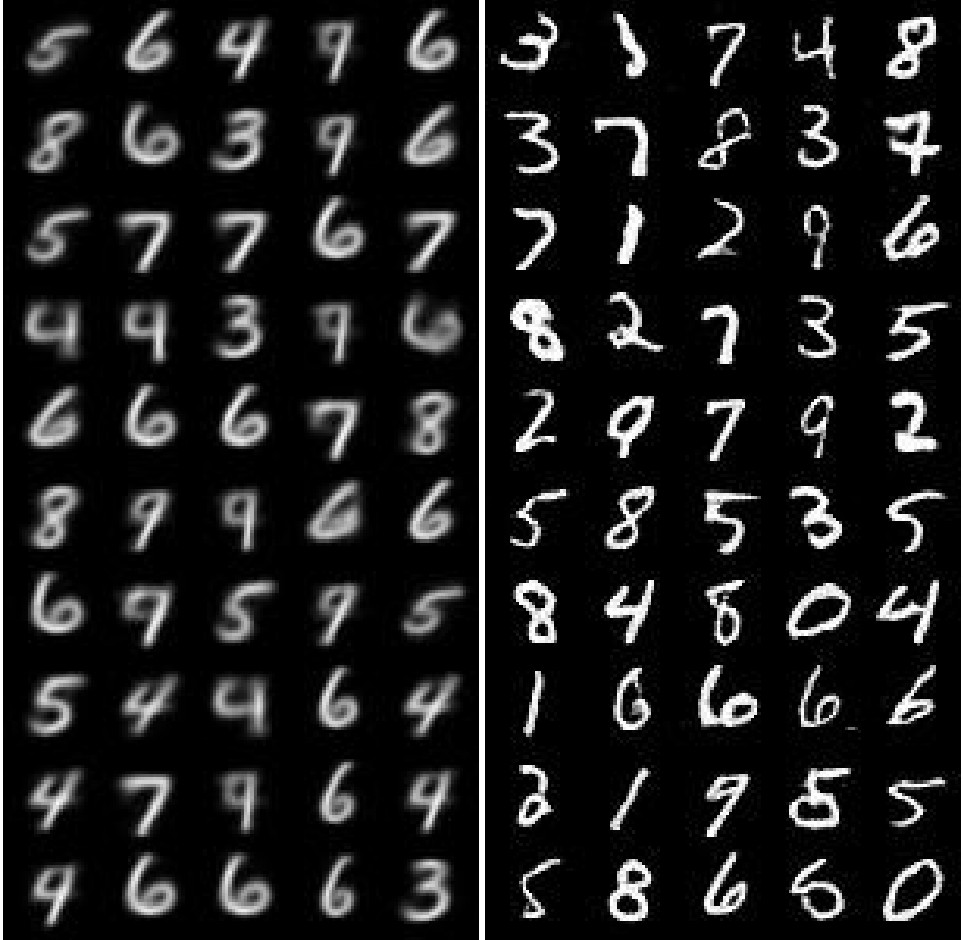

Figure 11: VAE (**left**) and GAN (**right**) samples with 200 latent variables. Each row represents a sample of a combination of 5 digits generated by the model.

