# OpenReview forum: "Parametric Adversarial Divergences are Good Task Losses for Generative Modeling"
_ICLR.cc/2018/Conference — Invite to Workshop Track_

### Official Review · AnonReviewer2 · 2017-11-21
**Interesting way to think about GANs**

**Rating:** 6
**Confidence:** 3

**Review:**

This paper is in some sense a "position paper," giving a framework for thinking about the loss functions implicitly used by the generator of GAN-type models. It advocates thinking about the loss in a way similar to how it is considered in structured prediction. It also proposes that approximating the dual formulation of various divergences with functions from a parametric class, as is typically done in GAN-type setups, is not only more tractable (computationally and in sample complexity) than the full nonparametric estimation, but also gives a better actual loss.

Overall, I like the argument here, and think that it is a useful framework for thinking about these things. My main concern is that the practical contribution on top of Liu et al. (2017) might be somewhat limited.

A few small points:

- f-divergences can actually be nonparametrically estimated purely from samples, e.g. with the k-nearest neighbor estimator of https://arxiv.org/abs/1411.2045, or (for certain f-divergences) the kernel density based estimator of https://arxiv.org/abs/1402.2966. These are unlikely to lead to a practical learning algorithm, but could be mentioned in Table 1.

- The discussion of MMD in the end of section 3.1 is a little off. MMD is fundamentally defined by the kernel choice; Dziugaite et al. (2015) only demonstrated that the Gaussian RBF kernel is a poor choice for MNIST modeling, while the samples of Li et al. (2015) simply by using a mixture of Gaussian kernels were much better. No reasonable fixed kernel is likely to yield good results on a harder image modeling problem, but that is a slightly different message than the one this paragraph conveys.

- It would be interesting to replicate the analysis of Danihelka et al. (2017) on the Thin-8 dataset. This might help clarify which of the undesirable effects observed in the VAE model here are due to likelihood, and which due to other aspects of VAEs (like the use of the lower bound).

---

> ### Author Response · Authors · 2018-01-03
> **Answer to AnonReviewer2.**
>
> We thank the reviewer for taking the time to review our paper, and for evaluating our paper as a position paper - which is indeed what we intended our paper to be.
>
> Concerning the difference with the work of Liu et al. (2017), we refer the reviewer to Shuang Liu's comment "Mathematical View vs. Philosophical View", as well as our comment "Difference with Liu et al. (2017) and Arora et al. (2017)". We have updated our related work section to better contrast our work with those works (see the revised version).
>
> The bottom line is that while Liu et al. (2017) concentrate more on the mathematical properties of parametric adversarial divergences, they do not attempt to study the meaning and practical properties of parametric divergences. In our paper, we start by introducing the notion of final task, which is our true goal, but is often difficult to formalize and hard to learn from directly. We then give arguments why parametric divergences can be good approximations/surrogates for the final task at hand. To do that, we review results from the literature, establish links with structured prediction theory, and perform a series of preliminary experiments to better understand parametric divergences by attempting to answer the following questions. How are they affected by various factors: discriminator family, transformations of the dataset? How important is the sample complexity? How good are they at dealing with challenging datasets such as high-dimensional data, or data with abstract structure and constraints?
>
> As you have noted, we are not claiming that we have a complete theory of parametric divergences. Rather, we are proposing new ways to think of parametric divergences, and more generally of the (final) task of generative modeling.
>
> We now answer the reviewer's questions:
>
> R: "f-divergences can actually be nonparametrically estimated purely from samples, e.g. with the k-nearest neighbor estimator of https://arxiv.org/abs/1411.2045, or (for certain f-divergences) the kernel density based estimator of https://arxiv.org/abs/1402.2966. These are unlikely to lead to a practical learning algorithm, but could be mentioned in Table 1."
>
> A: Thank you for pointing out that there is a rich literature on estimating f-divergences from samples. We have updated section 3.1 to include some of those techniques. However, one should note that those techniques all make additional (implicit or explicit) assumptions on the densities. We updated the table caption and Section 3.1 to reflect that.

---

> > ### Author Response · Authors · 2018-01-03
> > **part 2**
> >
> >
> > R: "The discussion of MMD in the end of section 3.1 is a little off. MMD is fundamentally defined by the kernel choice; Dziugaite et al. (2015) only demonstrated that the Gaussian RBF kernel is a poor choice for MNIST modeling, while the samples of Li et al. (2015) simply by using a mixture of Gaussian kernels were much better. No reasonable fixed kernel is likely to yield good results on a harder image modeling problem, but that is a slightly different message than the one this paragraph conveys."
> >
> > A: We updated section 3.1 to make it clear that MMD is fundamentally dependent on the choice of kernels. In particular we emphasize that the fact that MMD does not perform well for generative modeling is because generic kernels are used. We actually provide a more complete discussion of the choice of kernels in Section 3.2 "Ability to Integrate Desirable Properties for the Final Task", where we discuss the possibility to learn the kernel based on data instead of hand-defining it. That discussion was motivated by the possibility of integrating more knowledge about the final task into the kernel.
> >
> > R: "It would be interesting to replicate the analysis of Danihelka et al. (2017) on the Thin-8 dataset. This might help clarify which of the undesirable effects observed in the VAE model here are due to likelihood, and which due to other aspects of VAEs (like the use of the lower bound)."
> >
> > A: Thank you for this interesting experiment idea. Indeed RealNVP is attractive as a generative model for comparing maximum likelihood and parametric divergences because the likelihood can be evaluated explicitly. However, we think that such an experiment is currently out of the scope of the paper because it is quite non-trivial for the following reasons.
> >
> > One reason is that there are no obvious extensions of RealNVP to convolutional architectures, which are arguably the best architecture to deal efficiently with high-resolution images. However, there are other generators with also feature explicit likelihood. Probably one of the best known architectures with explicit likelihood for image generation is the PixelCNN (Van den Oord et al. (2016), https://arxiv.org/pdf/1601.06759.pdf). Training them using maximum-likelihood is not a problem because teacher-forcing is used, which allows to parallelize the process. However training them using a discriminator requires generating samples, which is extremely slow, because images have to be generated pixel after pixel.
> >
> >
> > We hope we have addressed the reviewer's concerns and we thank the reviewer again for their constructive review.

---

### Official Review · AnonReviewer1 · 2017-11-26
**A very general formulation without real theoretical or empirical support**

**Rating:** 4
**Confidence:** 4

**Review:**

This paper introduces a family of "parametric adversarial divergences" and argue that they have advantages over other divergences in generative modelling, specially for structured outputs.

There's clear value in having good inductive biases (e.g. expressed in the form of the discriminator architecture) when defining divergences for practical applications. However, I think that the paper would be much more valuable if its focus shifted from presenting a new notion of divergence to deep-diving into the effect of inductive biases and presenting more specific results (theoretical and / or empirical) in structured prediction or other problems.  In its current form the paper doesn't seem particularly strong for either the divergence or GAN literatures. Some reasons below:

* There are no specific results on properties of the divergences, or axioms that justify them. I think that presenting a very all-encompassing formulation without a strong foundation does not add value.
* There's abundant literature on f-divergences which show that there's a 1-1 relationship between divergences and optimal (Bayes) risks of classification problems (e.g. Reid at al. Information, Divergence and Risk for Binary Experiments in JMLR and Garcia-Garcia et al. Divergences and Risks for Multiclass Experiments in COLT).  This disproves the point that the authors make that it's not possible to encode information about the final task in the divergence. If the loss for the task is proper, then it's well known how to construct a divergence which coincides with the optimal risk.
* The divergences presented in this work are different from the above since the risk is minimised over a parametric class instead of over the whole set of integrable functions. However, practical estimators of f-divergences also reduce the optimization space (e.g. unit ball in a RKHS as in Nguyen et al.  Estimating Divergence Functionals and the
Likelihood Ratio by Convex Risk Minimization or Ruderman et al. Tighter Variational Representations of f-Divergences via Restriction to Probability Measures). So, given the lack of strong foundation for the formulation, "parametric adversarial divergences" feel more like estimators of other divergences than a relevant new family.
* There are many estimators for f-divergences (like the ones cited above and many others based e.g. on nearest-neighbors) that are sample-based and thus correspond to the "implicit" case that the authors discuss. They don't necessarily need to use the dual form. So table 1 and the first part of Section 3.1 are not accurate.
* The experiments are few and too specific, specially given that the paper presents a very general framework. The first experiment just shows that Wasserstein GANs don't perform well in an specific dataset and use that to validate a point about those GANs not being good for high dimensions due to their sample complexity. That feels like confirmation bias and also does not really say anything about the parametric adversarial GANs, which are the focus of the paper.

In summary, I like the authors idea to explore the restriction of the function class of dual representations to produce useful-in-practice divergences, but the paper feels a bit middle of the road. The theory is not strong and the experiments don't necessary support the intuitive claims made in the paper.

---

> ### Author Response · Authors · 2017-12-20
> **Answer to AnonReviewer1**
>
> We thank the reviewer for their long and thorough review.
>
> Before we start addressing the reviewer's concerns, we would like to make it clear that we are a position paper. We are not claiming to introduce a new family of divergences. Rather, we are giving the name of "parametric adversarial divergence" to the divergences which have been used recently in GANs, and attempting to better understand why they are good candidates for generative modeling.
>
> We now answer the reviewer's points:
>
> R: "There are no specific results on properties of the divergences, or axioms that justify them. I think that presenting a very all-encompassing formulation without a strong foundation does not add value."
>
> A: It's actually very hard to obtain theoretical results for our work. What we claim is that parametric divergences can be a good approximation of our final task, which in the case of generation, is to generate realistic and diverse samples. It is not something that can be easily evaluated or proved: it is notoriously difficult to mathematically define a perceptual loss, so it's not obvious how to prove rigorously that parametric divergences approximate the perceptual loss well, other than by looking at samples, or using meaningful but debatable proxies such as inception score.
>
> R: "There's abundant literature on f-divergences which show that there's a 1-1 relationship between divergences and optimal (Bayes) risks of classification problems (e.g. Reid at al. Information, Divergence and Risk for Binary Experiments in JMLR and Garcia-Garcia et al. Divergences and Risks for Multiclass Experiments in COLT).  This disproves the point that the authors make that it's not possible to encode information about the final task in the divergence. If the loss for the task is proper, then it's well known how to construct a divergence which coincides with the optimal risk."
>
> A: What you are referring to is the equivalence between computing a divergence and solving a classification problem. This is seen in GANs as the discriminator is solving a classification problem with the appropriate loss between two distributions p and q, the loss of which corresponds to the divergence between p and q. In fact, by choosing the appropriate losses one can recover any f-divergence and any IPM (it corresponds to choosing the Delta in equation 1 of our paper).
> However the binary loss here is very different from what we call task loss or final loss. The final loss is what we actually care about (images that respect perspective, that are not blurry, made of full objects). Instead the loss you are referring to is a loss that defines the binary classification problem between p and q. We updated the paper to include your references. Originally we were based on the work of Sriperumbudur et al 2012. Thank you for helping us complete the references.

---

> > ### Author Response · Authors · 2017-12-21
> > **part 2**
> >
> > R: " The divergences presented in this work are different from the above since the risk is minimised over a parametric class instead of over the whole set of integrable functions. However, practical estimators of f-divergences also reduce the optimization space (e.g. unit ball in a RKHS as in Nguyen et al.  Estimating Divergence Functionals and the
> > Likelihood Ratio by Convex Risk Minimization or Ruderman et al. Tighter Variational Representations of f-Divergences via Restriction to Probability Measures). So, given the lack of strong foundation for the formulation, "parametric adversarial divergences" feel more like estimators of other divergences than a relevant new family."
> >
> > A: Whether parametric divergences are a new family or simply estimators is more of an opinion. However, our opinion is that parametric divergences are a new family because they have very different sample complexities than their nonparametric counterparts, and because they will only match the moments that the discriminator family can represent.
> >
> > R: "There are many estimators for f-divergences (like the ones cited above and many others based e.g. on nearest-neighbors) that are sample-based and thus correspond to the "implicit" case that the authors discuss. They don't necessarily need to use the dual form. So table 1 and the first part of Section 3.1 are not accurate."
> >
> > A: This is true, thanks for pointing it out. However, all these methods make additional assumptions about the densities, some of which are conceptually similar to smoothing the density, which makes them different from the true f-divergence. We updated Section 3.1 to reflect that.
> >
> > R: The experiments are few and too specific, specially given that the paper presents a very general framework. The first experiment just shows that Wasserstein GANs don't perform well in an specific dataset and use that to validate a point about those GANs not being good for high dimensions due to their sample complexity. That feels like confirmation bias and also does not really say anything about the parametric adversarial GANs, which are the focus of the paper.
> >
> > A: For the first experiment [Sample Complexity], it is well known that models trained with parametric divergences have no trouble generating MNIST and CIFAR. See for instance the DCGAN paper (https://pdfs.semanticscholar.org/3575/6f711a97166df11202ebe46820a36704ae77.pdf) and the WGAN-GP paper (https://arxiv.org/pdf/1704.00028.pdf).
> > On the contrary, using the true Wasserstein yields bad results on CIFAR. Our point is to raise awareness that parametric Wasserstein is NOT nonparametric Wasserstein, by showing that the resulting samples are much worse.
> > The second experiment [Robustness to Transformations] focuses on understanding actual properties of parametric divergences, by seeing how robust they are to simple transformations such as rotations and additive noise.
> > The third and fourth experiment compare parametric divergences with nonparametric divergences by taking a popular parametric divergence: the parametric Wasserstein and comparing with the most popular nonparametric divergence: the KL. Using them to train exactly the same generator architectures, we see that KL fails as the resolution goes from 32x32 to 512x512, while the parametric Wasserstein yields results with comparable quality to the training set. Similarly on the task of generating sequence of 5 digits that sum to 25, we see that the parametric Wasserstein is better at enforcing the constraint than the KL.
> >
> > To sum up, our experiments show the difference between parametric and nonparametric divergence, study invariance properties of parametric divergences, and compare how well parametric and nonparametric divergences can deal with high-dimensionality and enforcing constraints.
> >
> > It's true we do not have strong theory. But as we stated in the beginning, it's very challenging to prove that parametric divergences are a good proxy for human perception, when mathematically defining human perception is itself challenging. So the best we can do is to study the properties of the parametric divergences.
> >
> > We hope we have addressed the reviewer's concerns and thank the reviewer again for their time.

---

### Official Review · AnonReviewer3 · 2017-12-03
**Limited novelty for GAN and adversarial divergences literature**

**Rating:** 4
**Confidence:** 3

**Review:**

This paper takes some steps in the direction of understanding  adversarial learning/GAN and relating GANs and structured prediction under statistical decision theory framework.

One of the main contribution of the paper is to  study/analyze parametric adversarial divergences and link it with structured losses. Although, I see a value in the idea considered in the paper, it is not clear to me how much novelty does this work bring on top of the following two papers:

1) S. Liu. Approximation and convergence properties of generative adversarial learning. In NIPS, 2017.
2) S. Arora. Generalization and equilibrium in generative adversarial nets (GANs). In ICML, 2017.

Most of their theoretical results  seems to be already existing in literature (Liu, Arora,  Arjovsky) in some form of other and it is claimed that this paper put these result in perspective in an attempt to provide a more principled view of the nature and usefulness of adversarial divergences, in comparison to traditional divergences.

However, it seems to me that the paper is limited both in theoretical novelty and practical usefulness of these results. Especially, I could not see any novel contribution for GAN literature or adversarial divergences.

I would suggests authors to clearly specify novelties and contrast their work with
1) GAN literature: ([2] Arora's results)
2)  Adversarial divergences literature: ([1] Liu)

Also, provide more experiments to support several claims (without any rigorous theoretical justifications) made in the paper.

---

> ### Author Response · Authors · 2018-01-03
> **Answer to AnonReviewer3**
>
> We thank the reviewer for taking the time to review our paper.
>
> We now answer the reviewer’s comments and questions.
>
> R: “Most of their theoretical results  seems to be already existing in literature (Liu, Arora,  Arjovsky) in some form of other and it is claimed that this paper put these result in perspective in an attempt to provide a more principled view of the nature and usefulness of adversarial divergences, in comparison to traditional divergences.”
>
> Concerning the difference with the work of Liu et al. (2017), we refer the reviewer to Shuang Liu's comment "Mathematical View vs. Philosophical View", as well as our comment "Difference with Liu et al. (2017) and Arora et al. (2017)". Concerning the difference with the work of Arora et al. (2017), we also refer the reviewer to our comment "Difference with Liu et al. (2017) and Arora et al. (2017)".
> We have updated our related work section to better contrast our work with those works.
>
> The bottom line is that those works focus on specific mathematical properties of parametric divergences. Arora et al. (2017) focus on statistical efficiency of parametric divergences. Liu et al. (2017) focus on topological properties of adversarial divergences and the mathematical interpretation of minimizing neural divergences (in a nutshell: matching moments).
>
> However, neither of those works attempts to study the meaning and practical properties of parametric divergences. In our paper, we start by introducing the notion of final task, which is our true goal, but is often difficult to formalize and hard to learn from directly. We then give arguments why parametric divergences can be good approximations/surrogates for the final task at hand. To do that, we review results from the literature, establish links with structured prediction theory, and perform a series of preliminary experiments to better understand parametric divergences by attempting to answer the following questions. How are they affected by various factors: discriminator family, transformations of the dataset? How important is the sample complexity? How good are they at dealing with challenging datasets such as high-dimensional data, or data with abstract structure and constraints?
>
> R: “However, it seems to me that the paper is limited both in theoretical novelty and practical usefulness of these results. Especially, I could not see any novel contribution for GAN literature or adversarial divergences.”
>
> A: Here are some potential contributions to the adversarial divergence literature:
> - it is often believed in the GAN literature that weaker losses (in the topological sense) are easier to learn than stronger losses. There has indeed been work in the adversarial divergence literature on the relative strength and convergence properties of adversarial divergences. However, to the best of our knowledge, there is no rigorous theory that explains why weaker losses are easier to learn. By relating adversarial divergences used in generative modeling with the task losses used in structured prediction, we put into perspective some theoretical results from structured prediction theory that actually show and quantify how the strength of the objective affects the ease of learning the model. Because those results are consistent with the intuition that weaker divergences are easier to learn, they give additional reasons to think that this intuition is correct.
>
> We take this opportunity to emphasize that it is highly non-trivial to derive a rigorous theory on quantifying which divergences are better for learning. Unlike structured prediction, where the task loss is also used for evaluating the learned model, there is no one good way of evaluating generative models yet. Because a rigorous theory should study the influence of minimizing a divergence on minimizing the evaluation metric, any theory that is derived on divergences can only be as meaningful as the evaluation metric considered.

---

> > ### Author Response · Authors · 2018-01-03
> > **part 2**
> >
> > Now, we give some of our potential contributions to the GAN literature, and more generally to the generative modeling literature:
> > - we give further experimental evidence that parametric divergences can be better than maximum-likelihood for modeling structured data. We consider two tasks: modeling high-dimensional 512x512 data lying on a low-dimensional manifold (Thin-8 dataset), and modeling data with high-level abstract structure/constraints (visual hyperplane task). On both those tasks, we show that to train the same generator, minimizing a WGAN-GP parametric divergence yields better samples than optimizing objectives related to maximum likelihood (VAE evidence lower-bound).
> > - in the GAN literature, parametric divergences are commonly referred to as "lower bounds" or "variational lower bounds" of their corresponding nonparametric divergences (see for instance the f-GAN paper by Nowozin et al. (2017), https://arxiv.org/pdf/1606.00709.pdf). We think that the terminology is misleading, and we show in this paper that parametric divergences are not to be thought merely as a lower-bound of the corresponding nonparametric divergences. First, statistics-wise, parametric divergences have been shown to have very different sample complexities than nonparametric divergences. Moreover, if the final goal is generative modeling, parametric divergences can be more meaningful objectives; they have been shown to only match the moments that the discriminator family is able to represent, which in image generation seems to be enough to generate visually appealing samples. On the contrary, most nonparametric divergences are strict and enforce matching all moments, which is unnecessarily constraining, and might actually make the objective harder to learn. Finally, we illustrated experimentally that those differences do matter, by showing that using an objective derived from the true (nonparametric) Wasserstein yields worse results than using a parametric Wasserstein in high dimensions.
> > - to the best of our knowledge, we have not found extensive studies in the GAN literature of the behavior of parametric divergences with respect to transformations of the distributions.  This is important because in GANs, those divergences are minimized using gradient descent. Thus a divergence suitable for generative modeling should vary smoothly with respect to sensible transformations of the dataset (such as deformations, for images) in order to provide a meaningful learning signal to the generator. Therefore, we carry out preliminary experiments to assess the invariance properties of some parametric divergences to simple transformations. One should note that although such simple transformations are not completely representative of the ones induced by a GAN during the course of learning, it is not obvious how to design more complex transformations, such as ones that depart from the data manifold (other than noise, or image blurring).
> >
> > Even if we are not yet capable of deriving a rigorous theory, we do believe that parametric divergences are strong candidates to consider in generative modeling, both as learning objectives and as evaluation metrics. As pointed out in Colin Raffel’s comment, our paper is laying some of the groundwork for designing more meaningful and practical objectives in generative modeling. We hope that our work helps other researchers get a better perspective on generative modeling, and acts as a reminder to always keep the final task, which is our true goal, in mind.
> >
> >
> > We hope we have addressed the reviewer's concerns and we thank the reviewer again for taking the time to review our paper.

---

### Public Comment · ~Shuang_Liu1 · 2017-12-20
**Mathematical View vs. Philosophical View**

I just want to compare this work with the following two papers:

(1) S. Liu et al. Approximation and convergence properties of generative adversarial learning. In NIPS, 2017.
(2) S. Arora et al. Generalization and equilibrium in generative adversarial nets (GANs). In ICML, 2017.

Specifically, I will compare this work with the "approximation" part of (1) and "generalization" part of (2).

(a) The "approximation" part of (1) basically shows the global minima of parametric adversarial divergences are those distributions that are indistinguishable from the target distribution under certain statistical tests.
(b) The "generalization" part of (2) basically shows the sample complexity scales polynomially with the number of parameters for parametric adversarial divergences, whereas the sample complexity is usually exponential or even infinite for non-parametric adversarial divergences.

Now, this work tries to understand (a) and (b) in a more philosophical way. The argument is as follows: humans do not need an exponentially large amount of samples to learn, therefore the loss function adopted by a human must be "parametric". Furthermore, the loss function adopted by humans are usually induced by a certain set of criteria. Therefore, using parametric adversarial divergences can both reduce the sample complexity and encourage prediction results to be more close to what humans would make.

---

### Author Response · Authors · 2017-12-21
**Difference with Liu et al. (2017) and Arora et al. (2017)**

We are posting this comment since two reviewers have asked us to clarify the difference between our work and:

(1) S. Liu et al. Approximation and convergence properties of generative adversarial learning. In NIPS, 2017.
(2) S. Arora et al. Generalization and equilibrium in generative adversarial nets (GANs). In ICML, 2017.

The following is an extract of our updated related work section.

Arora et al. (2017) argue that analyzing GANs with a nonparametric (optimal discriminator) view does not really make sense, because the usual nonparametric divergences considered have bad sample complexity. They also prove sample complexities for parametric divergences. Liu et al. (2017) prove under some conditions that globally minimizing a neural divergence is equivalent to matching all moments that can be represented within the discriminator family. They unify parametric divergences with nonparametric divergences and introduce the notion of strong and weak divergence. However, both those works do not attempt to study the meaning and practical properties of parametric divergences. In our work, we start by introducing the notion of final task, and then discuss why parametric divergences can be good task losses with respect to usual final tasks. We also perform experiments to determine properties of some parametric divergences, such as invariance, ability to enforce constraints and properties of interest, as well as the difference with their nonparametric counterparts. Finally, we unify structured prediction and generative modeling, which could give a new perspective to the community.

---

### Public Comment · ~Colin_Raffel1 · 2017-12-28
**My take on this paper**

I am writing this comment because I enjoyed this paper and was surprised to see that the reviews were low.

First and foremost, I read this paper as a "position paper" (as AnonReviewer2 noted) - that is, it is making a philosophical argument about how to evaluate generative models.  Specifically, it's arguing that using parametric adversarial divergences (e.g. the critic/discriminator objective functions used in GAN training) as an evaluation metric is an interesting and potentially useful idea.  The argument amounts to
1. Parametric adversarial divergences have good sample efficiency and are straightforward to compute compared to their non-parametric/non-adversarial counterparts (Table 1).
2. It's easy to integrate prior knowledge about what you want to measure via the design of the critic/discriminator architecture (as they write, "The form of the discriminator may determine what aspects the divergence will be sensitive or blind to.")
3. It's kind of like choosing a good "structured loss" for structured prediction.  Weakening the structured loss allows learning.
They back up these points with a few experiments, showing e.g. that a GAN discriminator can learn unusual characteristics of the task that a VAE can't.

I like this paper because it presents a strong and thorough argument in favor of considering adversarial divergences for generative modeling.  After reading the paper I was personally sold on the idea (at least to the extent that I'm interesting in seeing practical applications and implementations of it).  While this paper *relies* on some prior theory on GANs, the idea and goals of the paper are distinct from those works -- namely, this paper is trying to convince people to adopt a new way of thinking about generative modeling evaluation, not prove properties about adversarial divergences.  Separately, this paper does not provide a lot of practical advice, but I see this paper as a prerequisite to adopting a standardized way of using adversarial divergences for evaluation.  It provides a compelling argument that future work can refer to when proposing how to utilize these ideas in practice.

---

### Public Comment · ~Ilya_Tolstikhin1 · 2018-01-05
**Good overview paper with interesting new ideas**

I think this is a good overview paper. It nicely summarizes a very recent line of work related to the properties of the adversarial divergences: weaker parametric divergences ('neural' or 'adversarial' divergences) are much better suited for the goals of the unsupervised generative modeling than stronger non-parametric divergences. Even though this conclusion is not new and has been made in the literature before, the authors support it with yet another interesting argument originating in the theory of structured prediction.

I would say, the main novel contributions of this paper are as follows:
(1) A general view landing the generative modeling and the structure prediction tasks in one framework. In this paper the authors use this observation to conclude that one should prefer parametric divergences over the non-parametric ones when dealing with high-dimensional data in unsupervised tasks. But I do think potentially this new point of view may lead to many other interesting discoveries.
(2) I also like couple of experiments introduced in the paper, including the Thin-8 dataset and the 5-digit MNIST experiment. I am personally very curious to try out those in my future research. In particular, Thin-8 seems to demonstrate the "blurriness" effect of VAE much better than the 28x28 MNIST. I am curious if the authors are going to share the Thin-8 dataset?

---

> ### Author Response · Authors · 2018-01-09
> **Thin-8 and Visual Hyperplane will be released.**
>
> Thank you for your comment, Ilya.
>
> We will release the Thin-8 dataset as well as the Visual Hyperplane (MNIST digits summing to 25), as soon as our submission is de-anonymized, along with the data-augmentation code (elastic deformations for Thin-8).
>
> Please note that the Visual Hyperplane dataset is generated on-the-fly from MNIST: every time a sample is requested, a combination of 5 symbolic digits is sampled uniformly from all possible combinations that sum to 25, then a corresponding image is sampled from MNIST for each symbolic digit. Finally, the 5 images are concatenated.

---

> ### Author Response · Authors · 2018-04-11
> **Thin-8 Dataset**
>
> We have released the Thin-8 dataset (1585 samples, 16 people, 512x512 resolution).
>
> Please find it here: https://gabrielhuang.github.io/code/

---

### Public Comment · ~Carl-Johann_Simon-Gabriel1 · 2018-01-05
**Interesting high-level ideas with good experiments**

This work gives a nice overview of different generative modelling methods (mostly GAN- and VAE-variants). It makes an interesting link to structured learning (see Section 4.1), thereby enabling the future transfer of results and techniques from the vast supervised learning literature to the field of unsupervised learning. The authors for example note that the recent work of Osokin et al. (2017) in structured prediction could explain why some GAN-types are easier to train than others and why they generate perceptually better images (Section 4.3). We would like to emphasise that this paper does not delve into the mathematical implications of the link to structured prediction: it rather discusses them at a relatively high-level. The authors however test their hypotheses on a set of original and convincing experiments. They thereby introduce an interesting new dataset, which consists of high-resolution hand-written '8'-digits. Incidentally, they show that GANs can generate high resolution images very accurately when the intrinsic dimension of the data is low, which I find a very nice observation on its own.

Overall, I do understand that the rating of this paper is subject to controversy, but I think that even high-level reasonings that are not yet entirely formalised could benefit the general ML-community, especially when they are backed up by interesting empirical experiments.


Osokin, Bach, Lacoste-Julien, On structured prediction theory with calibrated convex surrogate losses, NIPS, 2017

---

### Decision · Program_Chairs · 2018-01-29
**ICLR 2018 Conference Acceptance Decision**

**Decision:**

Invite to Workshop Track

**Comment:**

Pros:
 - The paper proposes interesting new ideas on evaluating generative models.
 - Paper provides hints at interesting links between structural prediction and adversarial learning.
 - Authors propose a new dataset called Thin-8 to demonstrate the new ideas and argue that it is useful in general to study generative models.
 - The paper is well written and the authors have made a good attempt to update the paper after reviewer comments.

Cons:
- The proposed ideas are high level and the paper lack deeper analysis.
- Apart from demonstrating that the parametric divergences perform better than non-parametric divergences are interesting, but the reviewers think that practical importance of the results are weak in comparison to previous works.
With this analysis, the committee recommends this paper for workshop.